# Classification of Atypical White Blood Cells in Acute Myeloid Leukemia Using a Two-Stage Hybrid Model Based on Deep Convolutional Autoencoder and Deep Convolutional Neural Network

**DOI:** 10.3390/diagnostics13020196

**Published:** 2023-01-05

**Authors:** Tusneem A. Elhassan, Mohd Shafry Mohd Rahim, Mohd Hashim Siti Zaiton, Tan Tian Swee, Taqwa Ahmed Alhaj, Abdulalem Ali, Mahmoud Aljurf

**Affiliations:** 1School of Computing, Universiti Teknologi Malaysia, Johor Bahru 81310, Johor, Malaysia; 2King Faisal Specialist Hospital and Research Center, Riyadh 11564, Saudi Arabia; 3Bioinspired Device and Tissue Engineering Research Group, School of Biomedical Engineering and Health Sciences, Faculty of Engineering, Universiti Teknologi Malaysia, Skudai 81300, Johor, Malaysia; 4Faculty of Information Technology, City University, Petaling Jaya 46100, Selangor Darul Ehsan, Malaysia

**Keywords:** acute myeloid leukemia, atypical white blood cells, autoencoder, CNN, augmentation

## Abstract

Recent advancements in artificial intelligence (AI) have led to numerous medical discoveries. The field of computer vision (CV) for medical diagnosis has received particular attention. Using images of peripheral blood (PB) smears, CV has been utilized in hematology to detect acute leukemia (AL). Significant research has been undertaken in the area of AL diagnosis automation in order to deliver an accurate diagnosis. This study addresses the morphological classification of atypical white blood cells (WBCs), including immature WBCs and atypical lymphocytes, in acute myeloid leukemia (AML), as observed in peripheral blood (PB) smear images. The purpose of this work is to build a classification model for atypical AML WBCs based on their distinctive features. Using a hybrid model based on geometric transformation (GT) and a deep convolutional autoencoder (DCAE), this work provides a novel technique in the field of AI for resolving the issue of imbalanced distribution of WBCs in blood samples, nicknamed the “GT-DCAE WBC augmentation model”. In addition, to extract context-free atypical WBC features, this study develops a stable learning paradigm by incorporating WBC segmentation into deep learning. In order to classify atypical WBCs into eight distinct subgroups, a hybrid multiclassification model termed the “two-stage DCAE-CNN atypical WBC classification model” (DCAE-CNN) was developed. The model achieved an average accuracy of 97%, a sensitivity of 97%, and a precision of 98%. Overall and by class, the model’s discriminating abilities were exceptional, with an AUC of 99.7% and a class-wise range of 80% to 100%.

## 1. Introduction

AML is a fast-growing malignancy characterized by a rapid increase in the number of immature blood cells. These immature cells then proliferate, replace regular blood cells, and inhibit bone marrow from creating healthy cells [1]. Identifying immature WBCs is the first step in diagnosing AML. This technique is mostly predicated on classifying WBCs as immature or normal cells, and then classifying immature cells into subgroups. Due to the complexity and similarity of immature cells, their classification is a formidable challenge. Nevertheless, according to long-term clinical experience in this field, some intermediate stages of myelopoiesis are susceptible to misclassification, particularly for WBCs in subsequent maturation stages, such as myelocytes and metamyelocytes [2,3]. This is due to the complexity of the maturation phases and the minimal variation between classes of continuous stages. Consequently, no strict standards can be developed to distinguish WBCs at various developmental phases [4,5]. Furthermore, the low frequency of specific forms of WBCs in AML blood samples makes it challenging for ML models to learn significant features for distinguishing between different types of WBCs [6,7]. Manual identification of immature cells is laborious, time-consuming, and susceptible to inter- and intra-class variation. Furthermore, certain advanced microscopes use quantitative approaches rather than qualitative methods based on pattern recognition and computer vision, which results in reduced sensitivity to blast cells. Previous studies found that the agreement between Cellavision DM96 (an advanced type of microscope) and pathologists in diagnosing leukemia was just 76.6% [1,8]. Automated solutions based on computer vision were developed to address this issue. These systems employed both conventional ML and DL techniques. Traditional ML employs hand-crafted features, whereas DL employs more abstract, automated features. However, unlike automated features, hand-crafted features are manual, low-level, and have restrictions, such as the necessity for human-defined criteria that necessitate subject-matter expertise. Therefore, the goal of this study is to develop a new DL classification model for classifying atypical WBC into subtypes, including atypical lymphocytes, monoblasts, myelocytes, myeloblasts, promyelocytes, promyelocytes (bilobed), monoblasts, and erythroblasts. This study focuses on the classification of atypical white blood cells into different types. This research contributes the following to current knowledge:A new WBC augmentation model called “the GT-DCAE WBC augmentation model” is developed by combining a geometric transformation model and a generative model by using deep convolutional autoencoder.A new model for classifying atypical white blood cells (WBCs) that includes immature WBCs and atypical lymphocytes is created. This model is called “the Two-stage DCAE-CNN atypical WBC classification model”, and it uses a combination of a deep convolutional autoencoder and a convolutional neural network.The newly proposed model is a context-free generalized model that incorporates only features associated with WBCs and excludes other blood components.

The rest of this article has the following structure: In Section 2, related work is discussed. Section 3 discusses the dataset and study methods, which include algorithms for WBC augmentation and unusual WBC categorization. The experiment’s results are discussed in Section 4. Section 5 concludes the article.

## 2. Related Work

Researchers are becoming increasingly interested in incorporating artificial intelligence into medical imaging [9,10]. In peripheral blood and bone marrow smears, the differential count of white blood cells (WBCs), especially immature and atypical cells, is a crucial clinical hematology assessment [11]. Several researchers sought to categorize various forms of WBCs, such as immature and atypical lymphocytes, into several subtypes by using ML and DL. Traditional ML methods rely on manually produced features, whereas DL methods use high-level abstract and automated representations of the image data [12]. Several researchers have employed ML with custom-designed features to classify abnormal WBCs. Suryani et al. [13] developed a system for differentiating between ALL and AML M3 by using WBCs’ morphological features, such as the WBC area, nucleus ratio, and granule ratio. Their system achieved an accuracy of 83.65%. Furthermore, they used a backpropagation momentum method to differentiate between the AML M2 and AML M3 subtypes. They achieved 94.29% accuracy for cell-based classification and 75% accuracy for image-based classification. Wiharto et al. [14] utilized a sample of AML M0 and AML M1 to classify WBCs into myeloblasts, promyelocytes, and myelocytes with the K-nearest Neighbor (K-NN) by utilizing the WBC diameter and nuclear roundness features. They obtained a maximum accuracy of 67.28%. Later, Wiharto et al. [15] used a newly proposed classification system to improve the classification accuracy of blast cells—specifically, myeloblasts, promyelocytes, and myelocytes. The proposed system was divided into two stages; the first stage involved pre-processing, image segmentation, and feature extraction. The second stage involved using the synthetic minority oversampling technique (SMOTE) to solve the problem of imbalanced data. Their proposed systems had an overall accuracy of 89.6%. Harjoko et al. [16] classified WBCs into the M1, M2, and M3 AML subtypes using the Active Contour without Edge (ACWE) method and a backpropagation momentum artificial neural network (ANN). Six features were used to train the model: the cell area, perimeter, circularity, nucleus ratio, mean, and standard deviation. The proposed system achieved a segmentation accuracy of 83.789% and a classification accuracy of 93.569%. Roy et al. [17] used an ANN and an adaptive neuro-fuzzy inference system to develop a classification system for AML M0, M1, M2, M3, and M4. They trained their model on a dataset of 600 AML cases to predict these five types of AML based on four CBC parameters: leukocytes, hemoglobin, platelets, and blasts. The average MSEs for the ANN and the neuro-fuzzy inference system were 0.0433 and 0.2089, respectively. Rawat et al. [18] devised a classification model for separating ALL and AML subtypes in leukemic cells. In addition, the classification model was able to classify AML and ALL into subtypes. AML was classified into AML M2, AML M3, and AML M5, while ALL was subdivided into L1, L2, and L3. The classification process made use of geometrical, textural, and color features. An SVM was utilized to implement the classification process, whereas a genetic algorithm was utilized to pick and optimize features. The classification system achieved a classification accuracy of 98.5%. Dasariraju et al. [19] identified 16 features related to nucleus size and shape, elliptical features, and color feature to classify immature WBCs as monoblasts, myeloblasts, erythroblasts, and promyelocytes by using a random forest. Other immature WBCs were not included due to a lack of resources. The WBCs were first segmented, and then a binary classification model was utilized to distinguish between mature and immature WBCs, followed by a multiclassification model for immature WBCs. The binary classification model had 91.23% precision and 95.41% sensitivity. Using an SVM, Dincic et al. [5] investigated morphological, fractal, and textural features for the classification of WBCs into 12 unique subtypes, including mature and immature cells. Cell area, nucleus-to-cell ratio, nucleus solidity, fractal dimension, correlation, contrast, homogeneity, and energy were retrieved as the most important features. They attained an average classification precision of 80%. Several researchers, on the other hand, used a deep learning approach to classify immature WBCs into different subtypes. Qin et al. [20] classified fine-grained WBCs based on a PB smear into 40 distinct subtypes by using deep residual learning. The classification accuracy ranged from 37% to 89%. The myeloblast, promyelocyte, neutrophil (band), neutrophil (segmented), eosinophil, lymphocyte, and monocyte classification accuracies were 75%, 62%, 77%, 76%, 51%, 87%, and 64.9%, respectively. Matek et al. [2] used blood smear images to classify AML WBCs into 15 different kinds, including normal, immature, and atypical cells, by using a deep learning system based on a CNN algorithm. The literature indicates that the classification of atypical WBCs has mostly centered on normal WBCs and acute lymphoid leukemia (ALL). However, limited study has been undertaken on the classification of atypical WBCs in AML, particularly immature WBCs, due to a number of obstacles.

## 3. Materials and Methods

### 3.1. Dataset

This work utilized a single-cell morphological dataset (AML Cytomorphology LMU) of leukocytes from AML patients and non-malignant controls. The dataset consisted of 18,365 single-cell images identified by experts and acquired from peripheral blood smears of 100 AML patients and 100 controls between 2014 and 2017 at the Munich University Hospital. The collection was categorized into 15 distinct single-cell image categories. Four of these were leukemic cells, whereas the remaining eleven were healthy white blood cells. Seven of the eleven categories were adult leukocytes, while four were immature. Expert pathologists evaluated malignant and noncancerous WBCs based on an established morphological classification [21]. Figure 1 shows samples of the fifteen different types of WBCs presented in the dataset.

### 3.2. The Proposed Model

In this study, the DCAE-CNN deep learning model was proposed to classify atypical WBCs into eight distinct subclasses. The WBCs were first segmented by using the CMYK-Moment Localization-Feature Fusion Extraction framework proposed by Elhassan et al. [7]. These cells were recognized by their uneven distribution in blood samples. Therefore, a new augmentation method based on the GT and DCAE generative model, which was called GT-DCAE, was proposed to generate additional synthetic WBC images.The proposed method comprised of two stages: a binary classification model to differentiate between typical and atypical WBCs and a multiclassification model to classify atypical WBCs into eight subtypes. This model is a hybrid of a DCAE network and a CNN. It first transformed the image into a new representation by using DCAE, and then passed the new representation to a CNN model for additional feature extraction. The proposed model consisted of four phases: phase I: WBC augmentation, phase II: WBC encoding and feature extraction, phase III: two-stage atypical WBC classification, and phase IV: model evaluation. Figure 2 demonstrates the components of the two-stage DCAE-CNN classification model. The following is an explanation of the model architecture’s details.

#### 3.2.1. Phase I: WBC Augmentation

In this phase, the WBC images were first geometrically transformed by using random rotation at varying angles (0°, 365°), as well as vertical and horizontal flipping. Other augmentation techniques, such as zooming, shearing, and brightness adjustment, were avoided due to the sensitive nature of the problem, as utilizing these techniques could alter the properties of WBCs. The DCAE was then applied to the original and transformed images to generate new synthetic images. Figure 3 depicts the GT-DCAE WBC augmentation model. The following is a description of the GT-DCAE architecture.

The DCAE model was designed to generate new WBCs that were close to the original WBCs, but not identical. This model was trained to learn novel low-dimensional discriminative features of WBCs for image reconstruction with minimal errors by utilizing backpropagation and a distance loss function. The model consisted of three components: the encoder, the decoder, and the latent vector, which is also called the bottleneck. The encoders compressed the input data into a low-dimensional latent representation that the decoder utilized to reconstruct the original image. The latent vector is a collection of low-dimensional image representations that could be described as a collection of filtered images. Let *X* be the input WBC image, let *E* be the encoder function, let *D* be the decoder function, and let *Z* be the latent vector; the encoder, decoder, and loss functions can be defined as follows:(1)E:X→Z
(2)D:Z→X
(3)$$E,D=argmin_{E,D}||X-{EoD||}^{2}$$
where *EoD* is the predicted image. The following is a summary of the DCAE components.

**The encoder network**: Using filter banks, the encoder network performed several convolutional operations to generate a new set of feature maps. The encoder network comprised three convolutional layers of 32, 64, and 128 filters using a 3 × 3 kernel and a LeakyReLU activation function. Following every convolutional layer was a maximum pooling layer of size 2 × 2 and a one-step stride. This method yielded a collection of pooled feature maps with the greatest weights. In this situation, the maximum pooling layer could be viewed as a feature selection strategy analogous to the feature selection algorithms used in conventional ML approaches.**Latent vector space**: This was expressed as 28 × 28 × 128, with 28 × 28 being the image size and 128 representing the number of compressed feature mappings. To retain the semantics across the encoder and decoder units, we built a latent vector space by using convolutional layers as opposed to dense layers [22]. The latent vector could be obtained by using the following equation:
(4)Z=σ(X⊗W)+b
where *Z* denotes the latent vector, *X* is the WBC input image, and *W* and *b* are the weights and bias, respectively. σ denotes the activation function.**The decoder network**: This consisted of three convolutional layers of 128, 64, and 32 filters using a 3 × 3 kernel and a LeakyReLU activation function. To reconstruct the compressed image into the original, each convolutional layer was up-sampled by using a subsampling layer. The reconstruction process of the encoded image shown in Equation (Equation 1) can be expresses as follows:
(5)$$Y=\sigma (Z⊗W^{\prime })+C$$
where $W^{\prime }$ identifies the inverse operation across both weight dimensions of the *k*th feature map. *C* denotes the bias. Algorithm 1 shows the details of the GT-DCAE augmentation model.

**Algorithm 1** The GT-DCAE WBC augmentation algorithm.

**Input:**
     I(x, y, k) is the input image.     H is the image height.     W is the image width.     C is the number of channels.     F (x, y, k) is the convolutional filter.     P is the padding ϵ {valid, same}.     S is the striding window.     ψ is the activation function.     PS is the pooling size.     R is the dropout rate.     UPR is the upsampling rate.     X is the 1-D flattened image.     N is the number of neurons.     input_shape is the input image shape.     Encoder_output is the latent representation of the WBC image..**Function** conv (I,C,F,ψ,S,P) **Return** *transformed image*    $conv{\left(I,F\right)}_{x,y}=\sum_{m=1}^{H}\sum_{n=1}^{W}\sum_{k=1}^{C}F\left(m,n,k\right)I\left(x+m-1,y+n-1,k\right)$     $conv{\left(I,F\right)}_{x,y}=\psi conv{\left(I,F\right)}_{x,y}$
**EndFunction**
**Function** BN(I(x,y,k) **Return** *batch-normalized image*    $I_{Norm}=\frac{I-mean(I)}{std(I)}$    $I_{BN}=\gamma \,I_{Norm}+\beta$
**EndFunction**
**Function** Max-pool(I(x,y,k),S) **Return** *pooled_image*   $I_{Pool}=max_{i,j}^{PS}I\left(x+i,y+j\right)$
**EndFunction**
**Function** Upsampling ($I_{Pool},UPR$) **Return** *upsampled_image*.    $I_{UPR}={\left(I_{Pool}\right)}_{UPR}$
**EndFunction**
**Procedure** DropoutI(x,y,k),R   Freeze R neuron in CNNEndProcedure**Function** Dense(X,N,ψ) **Return** *transformed 1-D matrix*   $dense_{i}\left(X,units\right)=\sum_{j=1}^{N}w_{ij}X_{j}+\eta_{i}$    $dense_{i}\left(X,units\right)=\psi \left(dense_{i}\left(X,units\right)\right)$
**EndFunction**
**Function** GT I(x,y,k) **Return** *GT transformed image.*  GT=[]     
**for**

j=(1:15)

**do**
   **for** $\left(0^{o}:360^{o}\right)$ **do**     $I_{R}=Rotation\left(I,d\right)$     $I_{GT}=I_{GT}.append\left(I_{R}\right)$     $I_{V}=V.flip\left(I\right)$     $I_{H}=H.flip\left(I\right)$     $I_{GT}=I_{GT}.append\left(I_{V}\right)$     $I_{GT}=I_{GT}.append\left(I_{H}\right)$   **end for**
**end for**

**EndFunction**
**Function** Encoder (I, input_shape) **Return** *encoded_image.*    Eecoder_input = Input (input_shape)    Conv1 = $conv(I,C,f_{1},\psi ,S,P)$    Conv1 = Max-pool (Conv1, S)    Conv2 = $conv(I,C,f_{2},\psi ,S,P)$    Conv2 = Max-pool (Conv2, S)    Conv3 = $conv(I,C,f_{3},\psi ,S,P)$    Conv3 = Max-pool (Conv3, S)    Encoder_output = Conv3    Encoder = Model (Eecoder_input, Encoder_output)
**EndFunction**
**Function** Decoder (Encoder_output) **Return** * * decoded_image    Z = Encoder_output    Decoder_input = Input (input_shape)    Conv1 = $conv(Z,C,f_{3},\psi ,S,P)$    Conv1 = Upsampling (Conv1, S)    Conv2 = $conv(Conv1,C,f_{2},\psi ,S,P)$    Conv2 = Upsampling (Conv2, S)    Conv3 = $conv(Conv2,C,f_{1},\psi ,S,P)$    Conv3 = Upsampling (Conv3, S)    Decoder_output = Conv(Conv3,3,3,ψ,S,P)    Decoder = Model (Decoder_input, Decoder_output)
**EndFunction**



#### 3.2.2. Phase II: WBC Encoding and Feature Extraction

In this phase, the WBC images were first transformed into a new image representation (28 × 28 × 128) by using the DCAE–encoder unit, where 28 × 28 represents the size of the encoded image and 128 represents the number of retrieved features. The new image representation was then fed into the the two-stage DCAE-CNN classification mode.

#### 3.2.3. Phase III: The Two-Stage Atypical WBC Classification

This model consisted of two consecutive stages, the first of which was a binary classification model for classifying WBCs into typical WBC vs. atypical WBCs. The second was a multiclassification model for further classifying atypical WBCs into one of eight distinct subtypes. Figure 4 illustrates the two-stage DCAE-CNN classification model. The following is a detailed description of the proposed classification model.

Stage I: The Typical vs. Atypical WBC Binary Classification Model

The Stage I classifier was a CNN model designed in a depth-wise fashion to enable the learning of more complex nonlinear functions. The model comprised three convolutional layers and one fully connected layer. The CNN’s input layer received the compressed image representation and applied a sequence of convolutional operations according to the following equation:(6)$$H\left(Z\right)=\sum_{m=1}^{28}\sum_{n=1}^{28}\sum_{k=1}^{128}F\left(m,n,k\right)Z\left(Z_{x}+m-1,Z_{y}+n-1,k\right)$$
where *H*(*Z*) is the set of feature maps obtained by applying convolutional operations on the latent vector *Z* at the special location *Z = ($Z_{x},Z_{y}$)*, and F is the kernel defined between the input *Z* and *H*.

Figure 5 illustrates Stage I (typical vs. atypical binary classification model).

The model was made up of three batch-normalized convolutional layers, with each having 64 of (28 × 28 × 128), 128 of (28 × 28 × 64), and 256 of (28 × 28 × 64) feature maps, and it employed a 3 × 3 kernel, the ReLU activation function, and L2 regularization. Following each of the first two convolutional layers, a maximum pooling layer to reduce the image dimensions. This procedure yielded 128 of (7 × 7) feature maps. A 20% dropout approach was utilized to prevent overfitting, while zero-padding was used to maintain boundary information and continuous feature map reductions.

Stage II: The Atypical WBC Multiclassification Model

Stage II is a multiclassification scheme for classifying atypical WBCs into eight subtypes, as mentioned earlier. Figure 6 illustrates Stage II (atypical WBC multiclassification model).

The model consisted of four batch-normalized convolution layers and one fully connected layer. The model received an input feature with a shape of 28 × 28 × 128. The convolutional layers consisted of 32 of (28 × 28 × 128), 64 of (28 × 28 × 32), 128 of (28 × 28 × 64), and 256 of (28 × 28 × 128) feature maps, respectively, which were generated by using a 3 × 3 kernel, the ReLU activation function, and L2 regularization. Following the third and fourth convolutional layers, there was a maximum pooling layer with a size of 2. The maximum pooling process produced 256 of (7 × 7) feature maps, which were subsequently passed through a fully connected layer. Zero-padding was used to preserve the boundary information and consistent feature map dimensions, while 20% dropout was used to avoid model overfitting. After the convolutional layers was a fully connected layer of 50 neurons. This layer received the output of the previous convolutional layer after flattening to produce feature maps of a uniform size. Algorithm 2 depicts the details of the two-stage DCAE-CNN model.
**Algorithm 2** The two-Stage DCAE-CNN atypical WBC classification algorithm.**Input:**     I(x,y,k) is the input image.**Function** Typical$\dot{v}$sȦtypical I (x,y,k) **Return** *WBC_type*     Z = Encoder (I)            Conv1 = conv (Z, C, $f_{1}$, ψ, S, P)     Conv1 = BN (Conv1)     Conv1 = Max-pool (Conv1, S)     Conv1 = Dropout (Conv1, 0.2)     Conv2 = conv (Conv1, C, $f_{2}$, ψ, S, P)     Conv2 = BN (Conv1)     Conv2 = Max-pool (Conv2, S)     Conv2 = Dropout (Conv2, 0.2)     Conv3 = conv (Conv1, C, $f_{3}$, ψ, S, P)     Conv3 = BN (Conv1)     Conv3 = Dropout (Conv2, 0.2)     Conv3 = flatten (Conv3)     Dense1 = Dense (Conv3, 50)     Dense1 = Dropout (Dense1, 0.2)     Output = Dense (Dense1, 1, sigmoid)**EndFunction****Function** Atypical_WBC_subtype I (x,y,k) **Return** *Atypical_WBC_subtype*     Z = Encoder (I)     Conv1 = conv (Z, C, $f_{1}$, Ψ, S, P)     Conv1 = BN (Conv1)     Conv2 = conv (Conv1, C, $f_{2}$, ψ, S, P)     Conv2 = BN (Conv1)     Conv3 = conv (Conv1, C, $f_{3}$, ψ, S, P)     Conv3 = BN (Conv1)     Conv3 = Max-pool (Conv3, S)     Conv3 = Dropout (Conv2, 0.2)     Conv4 = conv (Conv3, C, $f_{4}$, ψ, S, P)     Conv4 = BN (Conv4)     Conv4 = Max-pool (Conv4, S)     Conv4 = Dropout (Conv4, 0.2)     Conv4 = flatten (Conv4)     Dense1 = Dense (Conv4, 50)     Dense1 = Dropout (Dense1, 0.2)     Output = Dense (Dense1, 8, softMax)**EndFunction****Function** Two_Stage_atypical WBC_subtype I (x,y,k) **Return** *Atypical_WBC_subtype*     Read input image I (x, y, k)     GT(I)     Encoder(I)     WBC_type = Typical.vs.Atypical (I)**if** WBC_type = “Typical” **then**        Exit**else**   Return $WBC\_subtype=Atypical_{W}BC\_subtype\left(I\right)$**end if**EndFunction

#### 3.2.4. Model Training

The dataset was split 80/20 for training and testing. Both models were trained with SGD (0.8 momentum, 0.00001 learning rate). In Stage I, one neuron and a sigmoid function were used to classify WBCs as normal or abnormal. The Stage II model classified WBCs into eight subtypes by using a dense layer of eight neurons and a SoftMax loss function. The model was built by utilizing an Intel® CoreTM i7-9750 h at 2.60 GHz 192 CPU with 16 GB of RAM and an NVIDIA GeForce RTX 2070 with a max-design. The algorithm was written in Python by using Keras and other image-processing libraries to extract handcrafted features.

#### 3.2.5. Phase III: Model Evaluation

The DCAE model’s performance was assessed by calculating the mean square error between the original and synthetic images (MSE). Overall accuracy, sensitivity, precision, specificity, and the area under the receiver characteristic curve (AUC) were used to evaluate the classification performance.

## 4. Results

The results for both the GT-DCAE augmentation model and the two-stage DCAE-CNN classification model are discussed in detail in this section.

### 4.1. WBC Augmentation

Eight distinct GT-DCAE models were generated for each distinct subtype of atypical WBCs, and 10,000 WBC images were obtained for each subtype. The MSE values for all models ranged from 0.001 to 0.005, indicating that the distance between the original image and the generated image was minimal and that the model was able to produce similar images. In addition, as depicted in Figure 7, the proposed model displayed excellent calibration for all atypical WBC types by using the training and validation loss. To evaluate the significance of the proposed GT-DCAE augmentation method, the GT-DCAE method was compared to the standard GT augmentation method by using the proposed two-stage DCAE-CNN classification model, as shown in Table 4. The classification model performed better with the GT-DCAE than with the GT augmentation method. The details are presented in the section that follows.

### 4.2. Stage 1: Typical vs. Atypical Binary Classification

In this stage, WBCs were classified as either typical or atypical. The new dataset had 38,715 (51%) normal WBCs compared to 37,035 (49%) atypical WBCs. Using training and validation learning curves, the model stability was tested by analyzing the model convergence. Figure 8a,b depict the training and validation learning curves with regard to the loss and overall accuracy. Figure 8 shows that the validation learning curve spiked at the 15th epoch, which was considered an early stage of the learning process. However, after 15 epochs, the model began to converge, and by 50 epochs, the results were more stable. The training of the model was extended to 100 epochs to ensure learning stability. After 50 epochs, the training and learning curves converged, indicating that the model had reached convergence.

To evaluate the performance of the model, the level of agreement between the model predictions and actual values was calculated. The level of agreement was measured by using precision and sensitivity. The model attained a sensitivity of 97.77% and a precision of 97.42%. The AUC was utilized to evaluate the model’s capacity to discriminate between typical and atypical WBCs. The model attained a 99.99% AUC. Figure 9 shows a graphical illustration of the ROC curve obtained by plotting the true-positive rate (TPR) versus the false-positive rate (FPR) with different thresholds.

### 4.3. Stage II: The Atypical WBC Multiclassification Model

At this point, the dataset contained the eight distinct types of WBCs mentioned in the previous section. Using the training and validation learning curves, the model stability was tested by analyzing the model convergence. Figure 10a,b depict the training and validation learning curves with regard to the loss and overall accuracy, respectively. Figure 10 shows that after 50 epochs, the training and validation learning curves began to converge; however, after 150 epochs, the learning curves remained unchanged, indicating that the model had converged. Model training was extended to 200 epochs to assure the learning stability.

The classification performance of the Stage II model was tested by comparing pathologist-generated ground-truth labels with the model’s predictions. Each model prediction was a vector of eight probabilities, $\pi =[\pi_{1},\pi_{2},\cdots ,\pi_{8}]$, where $\pi_{i}$ corresponds to the *i^th^* subclass and iϵ[0,1,⋯,7]. The Argmax function was used to identify the class with the highest predicted probability. The model was evaluated by using the precision, sensitivity, F-score, and AUC. Table 1 displays the class-wise precision and sensitivity. It demonstrates that the model performed quite well in categorizing atypical WBCs, particularly myeloblasts, which are the most essential cell type for diagnosing AML. In classifying myeloblasts, the model reached 99% sensitivity and 99% precision. Additionally, the model performed exceptionally well when classifying other blast cells, such as erythroblasts and monoblasts.

In terms of precision, both erythroblasts and monoblasts possessed the highest level of precision, which was 100%. Moreover, the sensitivity for these types was 94% for erythroblasts and 86% for monoblasts, which is also considered to be high. This showed that there were no false positives for these two categories of WBCs; yet, a fraction of these WBCs were not recognized, indicating that there were false negatives. Promyelocytes (bilobed) and metamyelocytes, on the other hand, displayed the poorest precision: 20% and 33%, respectively. However, in terms of sensitivity, promyelocytes (bilobed) achieved 100%, while metamyelocytes achieved 50%. This showed that promyelocytes (bilobed) were classified with a significant rate of false positives, but no false negatives. Figure 11 reveals that promyelocytes (bilobed) were commonly misclassified as myeloblasts, an earlier stage of promyelocyte, while metamyelocytes were misclassified as myelocytes, an earlier stage of the metamyelocyte stage. This could be because promyelocytes (bilobed) and metamyelocytes are in the intermediate stages of myelopoiesis, which is known to be a difficult task and is subject to misclassification, as opposed to erythroblasts and monoblasts, which are considered be early stages of myelopoiesis. Erythroblasts is the earliest stage of erythropoiesis, which gives rise to red blood cells (RBCs), whereas monoblasts are the earliest stage of monocytopoiesis, that grow into monocytes or macrophages [23].

In terms of sensitivity, the model demonstrated that lymphocytes (atypical) and promyelocytes (bilobed) had the maximum sensitivity, with both having a 100% sensitivity. As opposed to this, the precision for lymphocytes (atypical) was 50%, indicating that this class was overestimated with no false negatives and some false positives, which may have been caused by the testing dataset’s sparse images that were available for this class. The least sensitive cell types, however, were metamyelocytes and promyelocytes, which demonstrated 50% and 53% sensitivity with 33% and 67% precision, respectively. However, given that these two types represent two consecutive phases of myelopoiesis and, as a result, have relatively low sensitivity and precision, this was to be expected. The model achieved a sensitivity of 67%, 20%, 88%, and 33% and a precision of 53%, 100%, 78%, and 50% for promyelocytes, promyelocytes (bilobed), myelocytes, and metamyelocytes, respectively, Figure 12 shows the class-wise sensitivity and precision of atypical WBCs.

The model yielded superior outcomes to those of earlier research by Matek et al., with the exception of promyelocytes and promyelocytes (bilobed). For promyelocytes, the model improved its precision, but not its sensitivity, whereas for promyelocytes (bilobed), it attained 100% sensitivity, but was unable to enhance the precision. Figure 11 demonstrates that both types of promyelocytes (PMO and PMB) were largely misclassified as myeloblasts, with 3 of 12 promyelocytes being misclassified as myeloblasts and 3 of 4 bilobed promyelocytes being classified as myeloblasts. However, as stated previously, the intermediate phases of myelopoiesis are subject to misclassification, especially for WBCs that mature in the following stages [2,3,4]. Since promyelocytes and myeloblasts represent consecutive stages of the same myelopoiesis lineage, namely, the granulopoiesis lineage, misclassification may occur [24,25]. Morphologically, the only distinction between promyelocytes and myeloblasts is that the cytoplasm of promyelocytes contains azurophilic granules, while the cytoplasm of myeloblasts has neither granules nor vacuoles.

The proposed model also demonstrated improved performance in identifying atypical lymphocytes, a type of cell reactivated by viral, bacterial, or parasitic infection. In terms of the size and volume of cytoplasm, atypical lymphocytes are comparable to monocytes; nevertheless, their nuclei are more regular than those of monocytes. Therefore, the precise classification of atypical lymphocytes is a difficult task. Despite this, the model was able to improve the atypical lymphocyte classification accuracy with a sensitivity of 100% and a precision of 50% when compared to the results from Matek et al.

The F-score of the proposed model summarized both the sensitivity and precision, as presented in Table 2 and Figure 13. The cells with the highest scores were myeloblasts and erythroblasts, whereas promonocytes and metamyelocytes had the lowest. This was due to the distinct natures of erythroblasts and myeloblasts, which stem from two distinct myelopoiesis lineages. In addition, the large number of images provided for myeloblasts allowed the model to learn more accurate features and generate more accurate findings. Promonocytes and metamyelocytes, on the other hand, are in sequential stages of myelopoiesis and are, consequently, subject to misclassification.

Additionally, the model demonstrated excellent discrimination between the eight classes, with an average AUC of 99.7%. Figure 14 displays the class-wise and overall ROC curves, as well as the AUCs. The AUCs for blast cells, the most significant WBC types for identifying AML, were 100%, 100%, and 99.4% for erythroblasts, monoblasts, and myeloblasts, respectively. In addition, the model’s ability to distinguish atypical lymphocytes from other WBCs was 90%, which was a decent result given the restricted number of atypical lymphocytes in the dataset. Other immature WBCs exhibited AUC values of 99.7%, 95.9%, 80.5%, and 99.3% for metamyelocytes, myelocytes, promyelocytes (bilobed), and promyelocytes, respectively.

The effectiveness of our model was evaluated based on two factors: first, the significance of the features extracted by the DCAE, and second, the use of the GT-DCAE synthetic images. Consequently, our model was compared to the following models:CNN model employing GT-DCAE images without features extracted by the DCAE to evaluate the significance of the DCAE-extracted features, as shown in Table 3.DCAE-CNN on GT images, excluding synthetic images generated by the DCAE model, to examine the impact of synthetic images on improving the classification accuracy, as presented in Table 4.

Table 3 demonstrates that the proposed model outperformed the CNN model in terms of the class-wise and total accuracy, demonstrating the significance of DCAE-extracted features.

Table 4 demonstrates that the use of the DCAE synthetic images led to an increase in class-wise accuracy—particularly sensitivity—and overall accuracy. The results of the proposed model were compared to those of other studies that utilized different methodologies on the same dataset. Matek et al. [2], Dasariraju et al. [19], and Dinčić et al. [5] were the accessible studies. A summary of these studies is provided in the section on related work. Table 5 compares our findings to those of these studies.

## 5. Conclusions

In this study, a two-stage deep learning model was developed to classify atypical WBCs. These cells are difficult to characterize and have received little scientific attention. This study combined the DCAE and a CNN to extract more discriminant WBC features and present a new model for classifying atypical WBCs. This research proposes a new augmentation strategy by using a hybrid model of geometric transformation and a deep convolutional autoencoder to improve the model classification performance by addressing imbalanced WBC distribution. Finally, the model’s results were compared to the ground truth and were benchmarked against existing classification methods of atypical WBCs; the model showed superior performance.

## Figures and Tables

**Figure 1 diagnostics-13-00196-f001:**
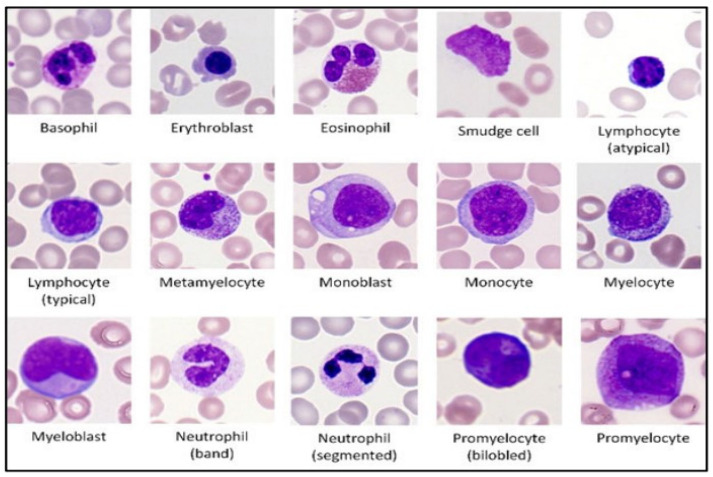
Samples of the fifteen different types of WBCs presented in the dataset, including normal and abnormal cells [5].

**Figure 2 diagnostics-13-00196-f002:**
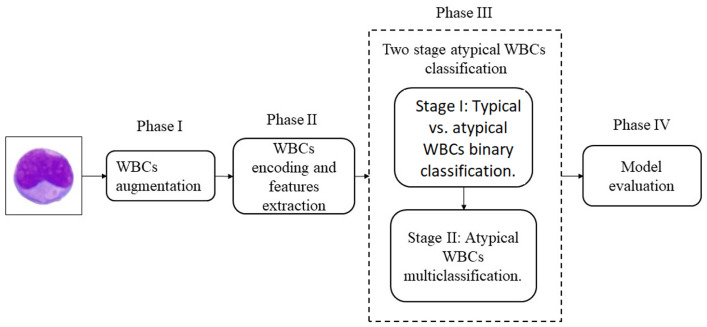
Components of the two-stage DCAE-CNN atypical WBC classification model.

**Figure 3 diagnostics-13-00196-f003:**
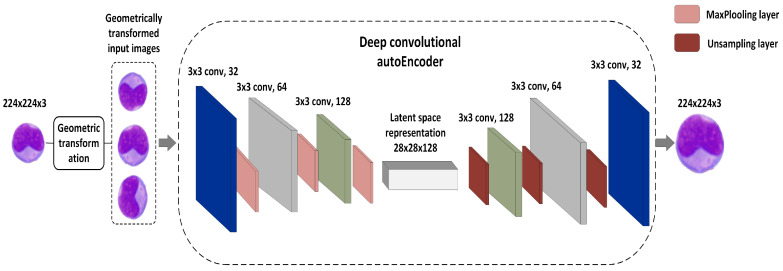
The GT-DCAE augmentation model.

**Figure 4 diagnostics-13-00196-f004:**
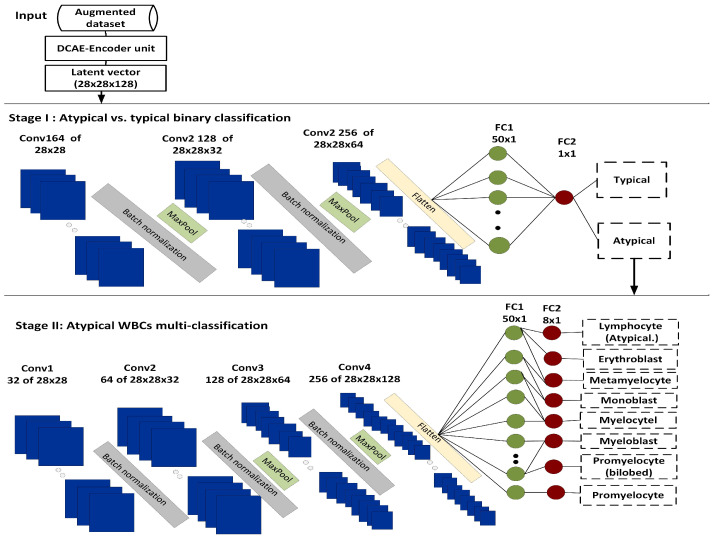
The two-stage DCAE-CNN atypical WBC classification model.

**Figure 5 diagnostics-13-00196-f005:**
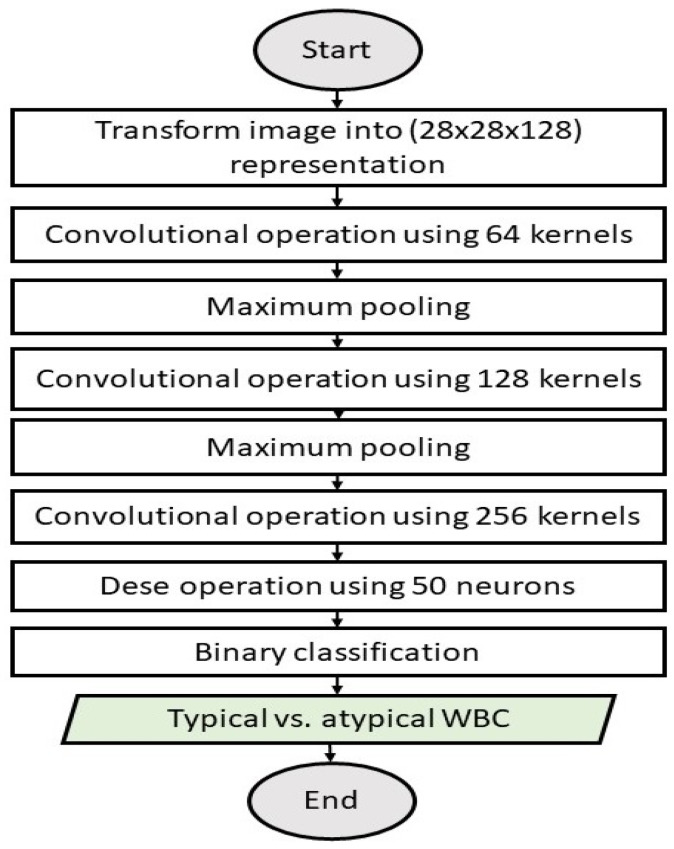
Stage I: Typical vs. atypical binary classification model.

**Figure 6 diagnostics-13-00196-f006:**
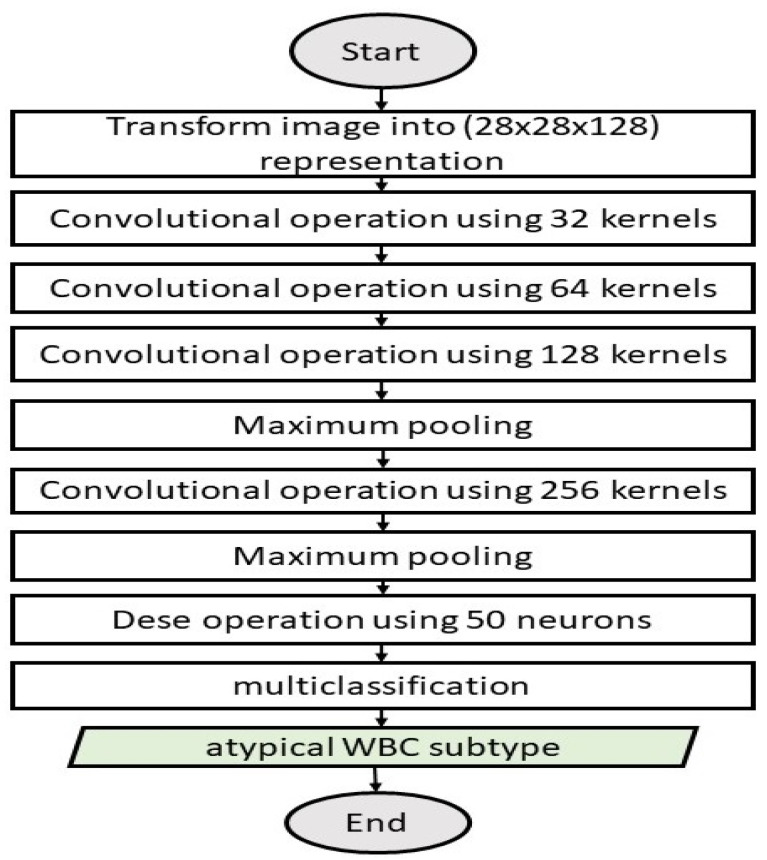
Stage II: Atypical WBC multiclassification model.

**Figure 7 diagnostics-13-00196-f007:**
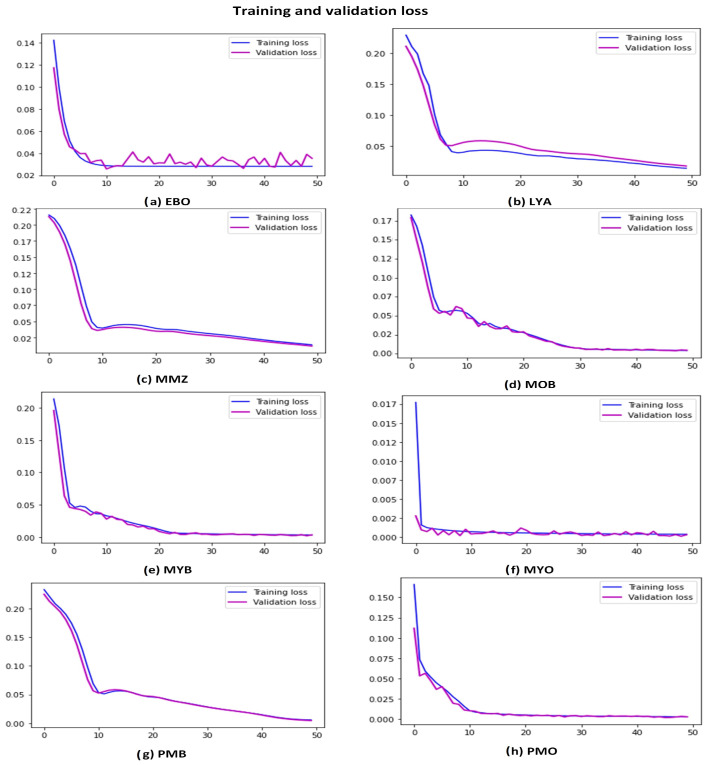
The DCAE model convergence based on training and validation loss.

**Figure 8 diagnostics-13-00196-f008:**
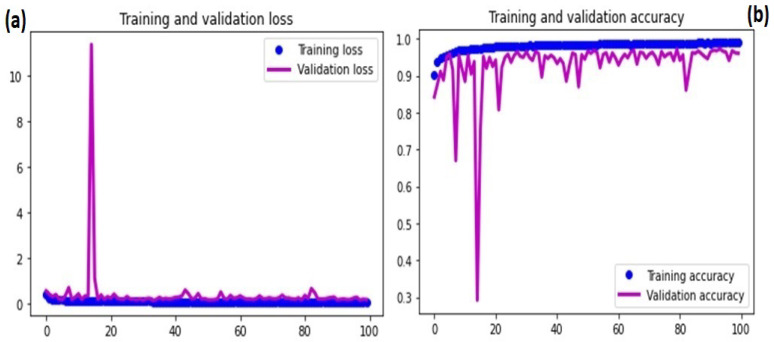
Stage I DCAE-CNN classification model convergence based on loss and accuracy: (**a**) training and validation loss; (**b**) training and validation accuracy.

**Figure 9 diagnostics-13-00196-f009:**
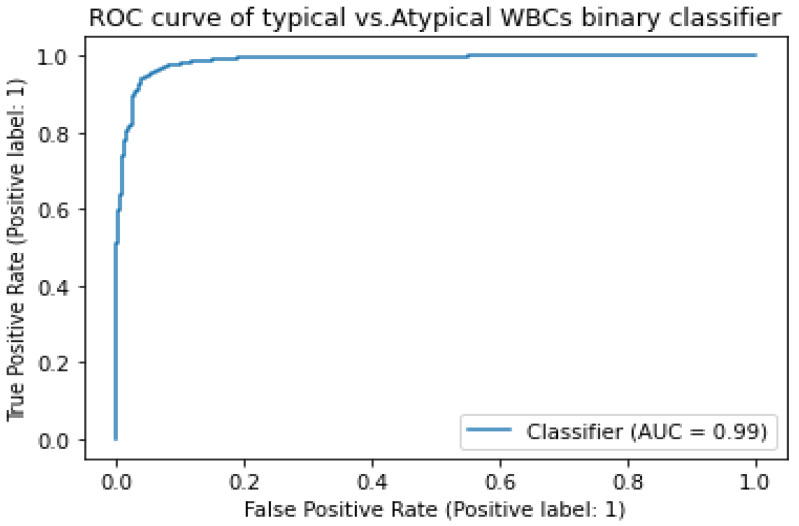
Stage I DCAE-CNN classification model ROC curve.

**Figure 10 diagnostics-13-00196-f010:**
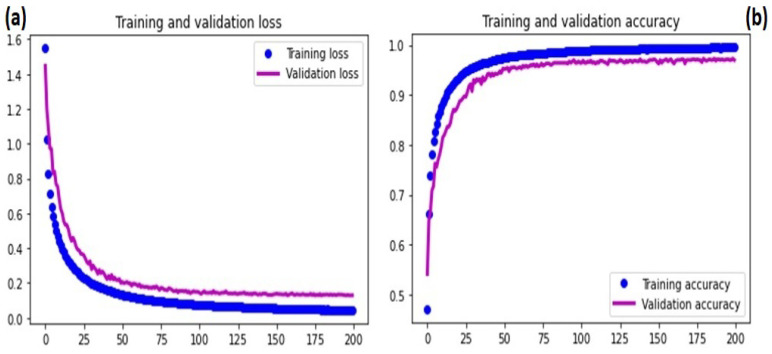
Stage II DCAE-CNN classification model convergence based on loss and accuracy: (**a**) training and validation loss; (**b**) training and validation accuracy.

**Figure 11 diagnostics-13-00196-f011:**
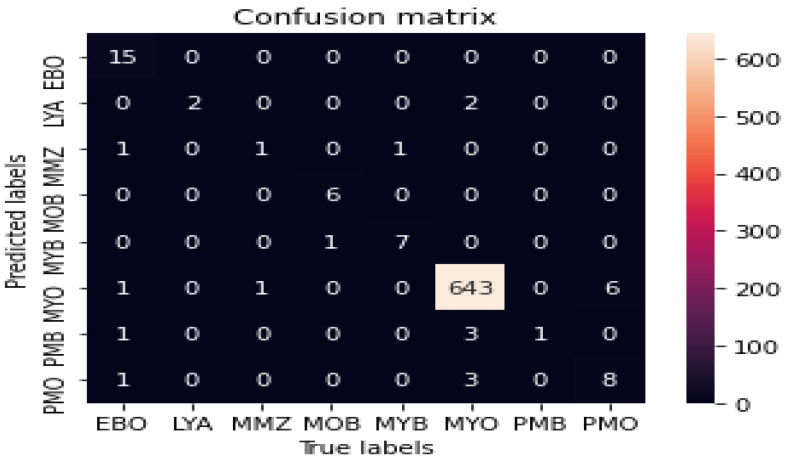
Confusion matrix for the atypical WBC multiclassification model.

**Figure 12 diagnostics-13-00196-f012:**
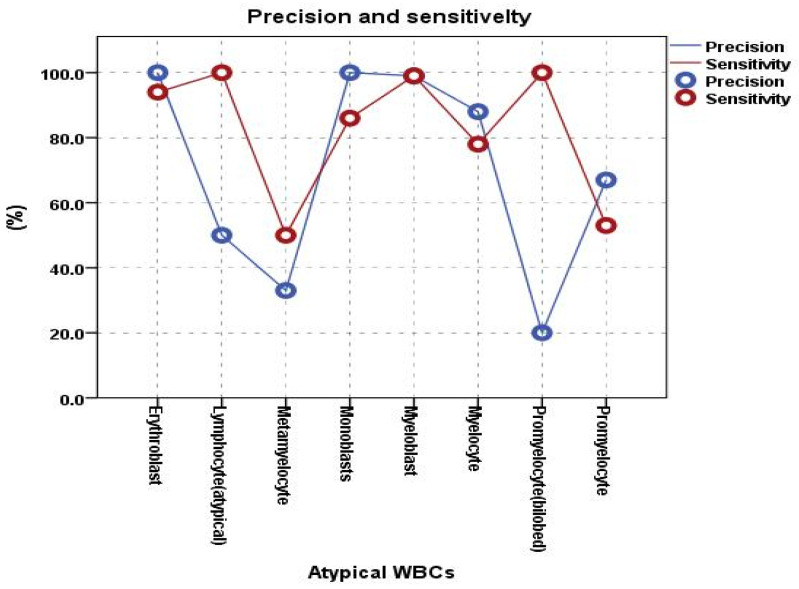
Precision and sensitivity of the Stage II DCAE-CNN.

**Figure 13 diagnostics-13-00196-f013:**
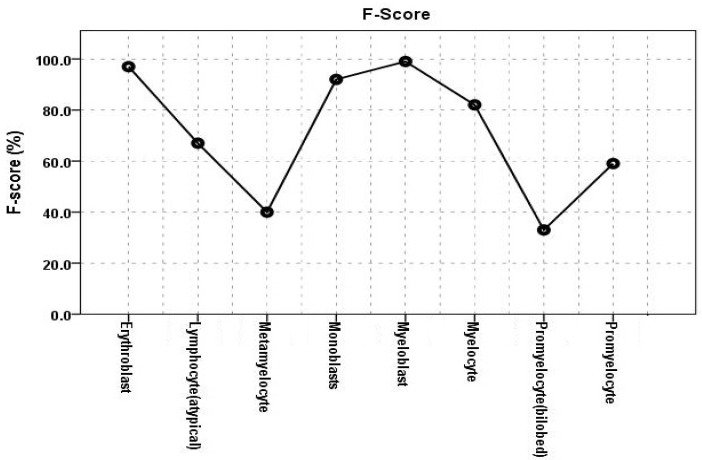
F-scores of Stage II: DCAE-CNN atypical WBC classification.

**Figure 14 diagnostics-13-00196-f014:**
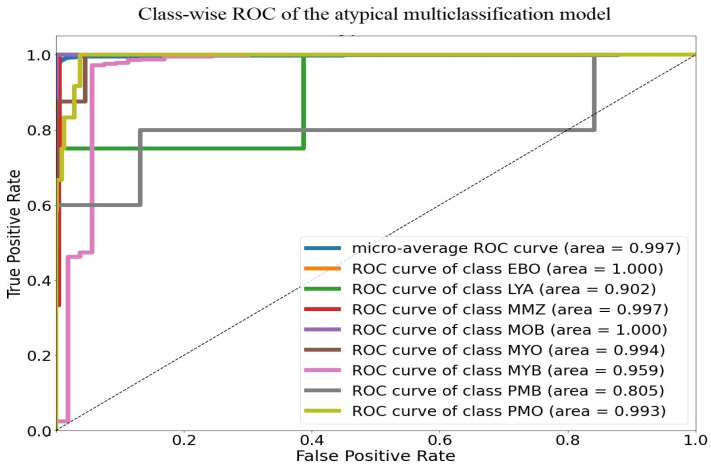
ROC curve for Stage II: DCAE-CNN classification.

**Table 1 diagnostics-13-00196-t001:** The classification results of the Stage II DCAE-CNN. of the DCAE-CNN in Stage II.

	Precision	Sensitivity	Number of Images/Class
Erythroblast	1.00	0.94	78.
Lymphocyte (atypical)	0.50	1.00	11
Metamyelocyte	0.33	0.50	15
Monoblast	1.00	0.86	26
Myeloblast	0.99	0.99	3268
Myelocyte	0.88	0.78	42
Promyelocyte (bilobed)	0.20	1.00	18
Promyelocyte	0.67	0.53	70

**Table 2 diagnostics-13-00196-t002:** F-scores and AUCs for r Stage II DCAE-CNN classification model.

WBCs	F-Score	AUC
Erythroblast	0.9700	1.0000
Lymphocyte (atypical)	0.6700	0.9020
Metamyelocyte	0.4000	0.9970
Monoblast	0.9200	1.0000
Myeloblast	0.9900	0.9900
Myelocyte	0.8200	0.9590
Promyelocyte (bilobed)	0.3300	0.80500
Promyelocyte	0.5900	0.9930

**Table 3 diagnostics-13-00196-t003:** DCAE-CNN and CNN Comparison of based on GT-DCAE augmented dataset.

	GT-DCAE		GT	
	**Precision**	**Sensitivity**	**Precision**	**Sensitivity**
Erythroblast	1.00	0.94	1.00	0.20
Lymphocyte (atyp)	5.00	1.00	0.00	0.00
Metamyelocyte	0.33	0.50	0.00	0.00
Monoblast	1.00	0.86	0.00	0.00
Myeloblast	0.99	0.99	0.93	0.90
Myelocyte	0.88	.78	0.00	0.00
Promyelocyte (bilobed)	0.20	1.00	0.00	0.00
Promyelocyte	0.67	0.53	0.00	0.00
Average overall accuracy	0.970		0.83	

**Table 4 diagnostics-13-00196-t004:** GT-DCAE and GT augmentation approaches based on a two-stage DCAE-CNN model comparison.

	GT-DCAE		GT	
	**Precision**	**Sensitivity**	**Precision**	**Sensitivity**
Erythroblast	1.00	0.94	1.00	0.79
Lymphocyte (atyp)	5.00	1.00	0.50	1.00
Metamyelocyte	0.50	1.00	0.50	1.00
Metamyelocyte	0.33	0.50	0.33	0.33
Monoblast	1.0	0.86	1.00	0.75
Myeloblast	0.99	0.99	0.95	1.00
Myelocyte	0.88	0.78	0.75	0.26
Promyelocyte (bilobed)	0.20	1.00	0.80	0.20
Promyelocyte	0.67	0.53	0.42	0.56
Promyelocyte	0.67	0.53	0.42	0.56
Average overall accuracy	0.97		0.93	

**Table 5 diagnostics-13-00196-t005:** Comparison with other models.

Authors	Matek et al. (2019) [2]		Dasariraju et al. (2020) [19]				Dincic et al. (2021) [5]		Our Model 2022			
**Prob.**	**Unadjusted**		**Unadjusted**		**Adjusted**		**Unadjusted**		**Unadjusted**		**Adjusted**	
**Metrics**	**Precision**	**Sensitivity**	**Precision**	**Sensitivity**	**Precision**	**Sensitivity**	**Precision**	**Sensitivity**	**Precision**	**Sensitivity**	**Precision**	**Sensitivity**
Erythroblast	0.7500	0.8700	1.0000	0.9130	0.9123	0.8710	0.8600	1.0000	1.0000	0.9400	0.9679	0.9303
Lymphocyte (atyp)	0.200	0.0700	-	-	-	-	-	-	0.5000	1.000	0.4839	0.9897
Metamyelocyte	0.070	0.1300	-	-	-	-	0.5000	0.4300	0.3300	0.5000	0.3194	0.4948
Monoblast	0.5200	0.5800	0.8750	1.0000	0.7982	0.9540	0.8800	0.9600	1.0000	0.8600	0.9679	0.8512
Myeloblast	0.9400	0.9400	0.9675	0.9444	0.8826	0.9009	0.8000	0.9600	0.9900	0.9900	0.9582	0.9798
Myelocyte	0.4600	0.4300	-	-	-	-	0.6500	0.5200	0.8800	0.7800	0.8517	0.7720
Promyelocyte (bilobed)	0.4500	0.4100	-	-	-	-	-	-	0.2000	1.0000	0.1935	0.9897
Promyelocyte	0.6300	0.5400	0.6250	0.8330	0.5701	0.5439	0.8900	0.7100	0.6700	0.5300	0.6484	0.5245
Overall Accuracy	-		0.9340		0.8676		0.8100		0.9700		0.9312	
AUC	0.9860		Not Cal		Not cal		Not cal		0.997		0.9897	

## Data Availability

The data are available at the following link: https://wiki.cancerimagingarchive.net/pages/viewpage.action?pageId=61080958 (accessed on 13 October 2022).

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
