# Peer review of "Classification of Atypical White Blood Cells in Acute Myeloid Leukemia Using a Two-Stage Hybrid Model Based on Deep Convolutional Autoencoder and Deep Convolutional Neural Network"

_diagnostics, 2023, doi:10.3390/diagnostics13020196_

Round 1

Reviewer 1 Report

The article is written well, easy to read and follow, and achieved promising results. I have a few comments and queries, though. First, I think there is no need to sectionize the abstract with background, methods, results, and conclusion. Second, the authors missed an important survey article on the topic and various SOTA methods listed both for conventional and deep learning models in "A Review on Traditional Machine Learning and Deep Learning Models for WBCs Classification in Blood Smear Images," which should be referred to and how their model compared to the techniques listed in the above survey achiving higher accuracies. This then raises questions on the significance and the novelty of the proposed model. For which I think the authors should, if not with other deep leanring models, at least compare results to conventional machine learning models, as they've used hand-crafted features for the atypical WBC. Additionally, I would like to ask how the results presented in Table 5 are even comparable with different numbers of classes. Did the author's test models proposed by Dasariraju et al. (2020) and Dincic et al. (2021) on their augmented dataset? If so, why didn't they report the results for Lymphocyte, promyelocyte, and other classes?

Other minor comments:

1.  add a full stop after Wiharto et al on line 78, 375

2. add space before reference [19] line 104, line 107, the start of a new sentence, line 145, line 150, line 226, 268, 330, 332, 373

3. remove s from learns on 164.

4. typo, hybrid, not hybris, on line 148

5. Remove The after Therefore, on line 144. 

6. Figure 2, Stage 1 in Phase III block states; Atypical vs atypical. I believe you mean typical vs atypical?

7. Improve the quality of Figure 3. 

8. Line 324, and 343, specify which studies the results are better than. Given a reference. 

9. A capital on line 375. 

10. Table 5, Our model 2022, "Udjusted" -- typo. 

Author Response

Response to Reviewer 1

Thank you for taking the time to review my manuscript. Indeed, your comments were very valuable and underlined the significance of the subject of our research. Following is point by point response to comments.

Point 1: think there is no need to sectionize the abstract with background, methods, results, and conclusion

Response 1: Thank you for bringing out the formatting issue with the abstract. The abstract has been revised into a single paragraph.

Point 2: the authors missed an important survey article on the topic and various SOTA methods listed both for conventional and deep learning models in "A Review on Traditional Machine Learning and Deep Learning Models for WBCs Classification in Blood Smear Images," which should be referred to and how their model compared to the techniques listed in the above survey achieving higher accuracies. This then raises questions on the significance and the novelty of the proposed model. For which I think the authors should, if not with other deep learning models, at least compare results to conventional machine learning models, as they've used hand-crafted features for the atypical WBC.

Response 2: Thank you for bringing up the survey article on leukocyte classification with TDM and DL. On line 68 of our manuscript, we cite the article. The review article, however, focusses on the classification of normal (mature) WBCs, which goes outside the scope of our study that focuses on the classification of atypical WBCs in acute myeloid leukemia (AML). Immature WBCs and atypical lymphocytes are examples of atypical WBCs. Atypical lymphocytes are frequently misclassified as blast cells, resulting in an AML diagnosis that is incorrect. Furthermore, in contrast to normal WBCs, atypical WBCs exhibit a high level of similarity and limited interclass variance, particularly in the subsequent phases of myelopoiesis, making classification task more challenging. Our classification methodology is comprised of two stages: the first step classifies WBC as typical (normal) versus atypical. If the result of the classification is "normal WBC," no further classification is required; otherwise, the WBC is further classified into eight distinct subtypes. Our results cannot be compared to those of the survey since the survey focused on the classification of mature, normal WBCs with different morphology, whereas our classification focuses on atypical WBCs with similar morphology that are not well separated.

Point 3: Additionally, I would like to ask how the results presented in Table 5 are even comparable with different numbers of classes.

Response 3: We compared our results to three authors who use the same public dataset, which contains 15 different subtype of WBCs including mature (normal), immature, blast, and atypical lymphocyte WBCs, as follows:

  1. Matek et al. (2019): Classified WBCs into 15 different types including atypical and typical WBCs. They also differentiate between typical versus atypical WBCs with an AUC of 0.991.

However, as our research focus on the classification of atypical WBCs, we developed a two-stage model, stage 1 is a binary classification model to classify WBC into typical vs. atypical, followed by stage 2 which a multi-classification model to classify atypical WBC into eight atypical subtypes. However, to compare our results with Matek et al. we adjusted our results by multiplying the class-wise sensitivity and precision by the sensitivity and precision of the typical versus atypical classification model. This is because the results of the multi-classification model depends on the accuracy of the binary classification model.  Table 5 contains the undusted and the adjusted probabilities of the class-wise probabilities.

  1. Dasariraju et al. (2020): Focused only on immature WBCS, therefore, they developed a classification model to classify immature WBCs into only four different subtypes including Monoblasts, Myeloblasts, erythroblasts, and Promyelocytes. They first performed binary classification to classify WBCs into mature vs. immature WBCs followed by a multi-classification model to classify immature WBCs into four distinct subtypes using random Forest. We followed the same strategy to calculate the class-wise adjusted probabilities obtained by the multi-classification model to benchmark their results with Matek et al. and with our results.

  1. Dincic et al. (2021): Focused on 11 WBCs subtypes based on handcrafted features and a multi classification SVM model. Therefore, as we did with Matek et al. to compared our adjusted results to their multi-classification results.

Point 4: Did the author's test models proposed by Dasariraju et al. (2020) and Dincic et al. (2021) on their augmented dataset? If so, why didn't they report the results for Lymphocyte, promyelocyte, and other classes?

Response 4: Our proposed augmentation model was designed to generate data that is suitable for deep learning. For the deep learning multi-classification task, the augmentation model generated 80,000 images. These kind of data, however, do not lend themselves to manual feature extraction or traditional machine learning algorithms. Machine learning requires less data. Furthermore, the authors compared their findings to Dasariraju (2020) and Dincic (2021) because they utilized the same dataset and focused on immature WBCs in AML, which is comparable to our work. Authors are welling for any further comments or clarification.

Point 5:   add a full stop after Wiharto et al on line 78, 375

Response 5: corrections are done and reflected in the manuscript.

Point 6:    add space before reference [19] line 104, line 107, the start of a new sentence, line 145, line 150, line 226, 268, 330, 332, 373

Response 6: corrections are done and reflected in the manuscript.

Point 7:  remove s from learns on 164.

Response 7: corrections are done and reflected in the manuscript.

Point 8:    typo, hybrid, not hybris, on line 148

Response 8: corrections are done and reflected in the manuscript.

Point 9:   Remove The after Therefore, on line 144.

Response 9: corrections are done and reflected in the manuscript.

 Point 10:   6. Figure 2, Stage 1 in Phase III block states; Atypical vs atypical. I believe you mean typical vs atypical?

Response 10: Figure 2 has corrected and reflected in the manuscript.

Point 11: Improve the quality of Figure 3. 

Response 11: the quality of figure 3 has been improved and reflected in the manuscript.

Point 12:  Line 324, and 343, specify which studies the results are better than. Given a reference. 

Response 12: corrections are done and reflected in the manuscript.

Point 13: A capital on line 375. 

Response 13: corrections are done and reflected in the manuscript.

Point 14: Table 5, Our model 2022, "Udjusted" -- typo. 

Response 14: corrections are done and reflected in the manuscript.

Reviewer 2 Report

The manuscript can be a useful tool for the diagnosis and treatment of patients with acute leukemia. The authors developed a two-stage deep learning model to classify atypical WBCs that are difficult to characterize. The study combined DCAE and CNN, proposed a new extension strategy using a hybrid geometric transformation model and a deep convolutional autoencoder to improve model classification performance by addressing the problem of unbalanced WBC decomposition. Finally, the model’s results were compared to ground truths and benchmarked against existing classification methods of atypical WBCs. The model achieved an average of 97% accuracy, 97% sensitivity and 98% precision.

Before publication, it is worth to standardize the references, eg in item 3 there are "Vol" and "pp" abbreviations, while in item 2 there are no such abbreviations.

Author Response

Response to Reviewer 2 Comments

Thank you for taking the time to review my manuscript. Indeed, your comments were very valuable and underlined the significance of the subject of our research. Following is point by point response to comments.

Point 1: Before publication, it is worth to standardize the references, e.g. in item 3 there are "Vol" and "pp" abbreviations, while in item 2 there are no such abbreviations.

Response 1: references have been standardized.